# In Vitro Gut Fermentation of Whey Protein Hydrolysate: An Evaluation of Its Potential Modulation on Infant Gut Microbiome

**DOI:** 10.3390/nu14071374

**Published:** 2022-03-25

**Authors:** Chunsong Feng, Li Tian, Hui Hong, Quanyu Wang, Xin Zhan, Yongkang Luo, Yuqing Tan

**Affiliations:** 1Beijing Laboratory for Food Quality and Safety, College of Food Science and Nutritional Engineering, China Agricultural University, Beijing 100083, China; s20203060975@cau.edu.cn (C.F.); tianli_cau@163.com (L.T.); hhong@cau.edu.cn (H.H.); luoyongkang@cau.edu.cn (Y.L.); 2Department of Product and Development, Heibei Dongkang Dairy Co., Ltd., Shijiazhuang 052165, China; quanyuwang@gmail.com (Q.W.); zxhrbcn@163.com (X.Z.)

**Keywords:** whey protein hydrolysate, fecal fermentation, infant gut microbiome, prebiotics

## Abstract

Whey protein and its hydrolysate are ubiquitously consumed as nutritional supplements. This study aimed to evaluate the potential effect of whey protein hydrolysate (WPH) on the infant gut microbiome, which is more variable than that of adults. Colonic fermentation was simulated through a static digestion model and fecal culture fermentation, using feces from normal infants aged from 1–3 years old. During in vitro gut fermentation, measurements of short-chain fatty acids (SCFA) concentrations and 16S rRNA amplicon sequencing were performed. Additionally, the growth curves of cultivated probiotics were analyzed to evaluate the prebiotic potential of WPH. Besides the decline of pH in fermentation, the addition of WPH induced a significant increase in the SCFA production and also the relative abundance of Proteobacteria, *Bacteroides*, and *Streptococcus* (*p* < 0.05). The lower ratio of Firmicutes/Bacteroidetes in WPH-supplemented samples indicated the positive modulation of WPH on the gut microbiota, which could benefit the energy balance and metabolism of infants. The stimulation effect of WPH on the probiotics (particularly *Lactobacillus acidophilus* NCFM) during cultivation implied the prebiotic potential as well. Our findings shed light on WPH as a valuable dietary supplement with not only enriched resources of essential amino acids but also the potential to restore the infant gut microbiome.

## 1. Introduction

As the supernatant by-product left after coagulation of casein in the dairy industry, the whey protein (WP) fraction represents approximately 15–20% of all protein in bovine milk and has been discovered to be an excellent source of food-grade protein [1]. The peptides released during enzymatic hydrolysis also enable WP to present many in vivo biological functions, such as benefits on the nervous, endocrine, immune, cardiovascular, and digestive systems [2]. As an enriched source of essential amino acids and other nutrients [3], WP is now commonly applied in infant formula (IF), which has been ubiquitously used as a substitute for breastfeeding. Due to the high prevalence of allergies to cows’ milk in infants, whey protein hydrolysate has been commercially developed as a major kind of hypoallergenic dairy product [4]. Moreover, “predigested” IFs seem to induce a better feeding tolerance for preterm infants, which was indicated by higher gastrointestinal transport rates, more rapid gastric emptying, less gastro-esophageal reflux, and more rapid establishment of full enteral feeding [5]. More recently, an IF with partially hydrolyzed WP was proved to support adequate growth for healthy term infants [6]. WP hydrolysate was also found to serve as a probiotic growth and metabolism enhancer for Wistar rats, which were fed with a standard diet [7]. However, few studies have been conducted to evaluate the potential effect of hydrolyzed WP on the infant gut microbiome.

The microbiota in human guts was affected by copious factors such as delivery pattern, diet, genetics, antibiotics, and prebiotics [8]. During the period of approximately three to five years after birth, the gut microbiota of individuals could have swift changes in microbial configuration till the relatively steady stage in adulthood. Those early-life occurrences even have a long-term impact on the brain and behavior [9]. The introduction of IF early in the postnatal period also shows interference in the colonization and proliferation of the neonatal intestinal microbiota, even given in small amounts during breastfeeding [10]. To properly modulate the microbiome, many attempts have focused on the application of prebiotics, which preferentially stimulates the growth and metabolic activity of probiotics. In general, prebiotics are chiefly thought to be various types of carbohydrates that are resistant to the digestion of the host [11]. However, some fragments from bovine and human breast milk were also reported to generate beneficial modulation of the gut microbiota [12]. WP-derived peptides could not only present bacteriostasis against pathogenic strains of *E. coli, Bacillus subtilis,* and *Staphylococcus aureus*, but also show growth-promoting effects on *Bifidobacterium longum* ATCC 15,707 [4]. Additionally, WP-derived peptides have also been associated with the improvement of oxidative stress and the maintenance of the intestinal barrier, such as a commercial WP-derived product named Hilmar™ 8350 [13].

In the current study, we sought to evaluate the potential modulation of the Hilmar™ whey protein hydrolysate (WPH) on the infant gut microbiome, using infant fecal inoculum and the cultivation of five probiotic strains. As a valuable reference for further nutritional supplement applications, information on the components of amino acids and peptides in WPH were obtained using HPLC-MS/MS.

## 2. Materials and Methods

### 2.1. Materials

Hilmar™ whey protein hydrolysate 8350 (WPH) was donated by Tianjin Yinhe Weiye Import & Export Co., Ltd. (Tianjin, China). A total of five species of probiotics were cultivated. *Lactobacillus rhamnosus* GG (LGG) and *Bifidobacterium animalis* BB-12, which were both isolated from milk, were kindly supplied by the microbiology laboratory in the College of Food Science and Nutritional Engineering, China Agricultural University (Beijing, China). *Lactobacillus acidophilus* NCFM was ordered from the China Center of Industrial Culture Collection (Beijing, China). *Bifidobacterium lactis* HN019 and *Bifidobacterium lactis* Bi-07 were obtained from DuPont™ Danisco (Shanghai, China). The digestive enzymes including pepsin and pancreatin were purchased from Sigma-Aldrich (St. Louis, MO, USA). Bovine bile salts were purchased from Solarbio (Beijing, China).

### 2.2. Determination of Amino Acids and Peptides in WPH

High-performance liquid chromatography with tandem mass spectrometry (HPLC-MS/MS) analysis was used to detect peptides in WPH. In short, analysis of amino acid composition followed the methods described by Liang et al. [14]. Peptides were determined according to Long et al. [15]. Peptide profiles with intensity values were visualized through Peptigram [16].

### 2.3. In Vitro Anaerobic Culture

*Lactobacilli* and *Bifidobacteria* were cultured as described by Li et al. [17]. The basal medium was supplemented with 3% and 5% WPH (occupation in nitrogen source) in two groups, respectively. The biomass was monitored spectrophotometrically every 2 h during 24-h culture, using OD_600_ measurements.

### 2.4. Simulated Gastrointestinal Digestion of WPH

The in vitro digestion of WPH was modified from the description of Brodkorb et al. [18]. Samples were incubated on a reciprocating shaker at 37 °C during digestion. Briefly, the digestion model contained a gastric phase (2 h) and an intestinal phase (2 h), where WPH was treated, respectively, with pepsin and pancreatin. Bovine bile salts were added into the gastric chyme before the start of the intestinal phase. A water bathing at 95 °C for 15 min was carried out to inactivate enzymes. The digestive product of WPH was submitted to freeze-drying treatment and then designated as DWP.

### 2.5. Determination of Molecular Weight Distribution

Molecular weight distribution profiles of WPH and DWP were determined by size exclusion high-performance liquid chromatography as described by Wang et al. [19].

### 2.6. Infant Fecal Inoculum

The infant feces were collected from 10 local babies (age range: 1–3 years old, location: Beijing) and were immediately stored at −80 °C until further use (in the presence of glycerol). The same amount of collected feces from each baby was mixed together before the inoculation. Those babies were fed or partially fed with infant formula of stage 3. There was no antibiotic treatment for both babies and their mothers during the last half year before feces sample collection. The anaerobic fermentation medium was prepared as described by Chen et al. [20] with slight modifications. The addition of nutritional agents to the 9 mL medium per tube was 100 mg of DWP in the low-dose group (L), 200 mg of DWP in the high-dose group (H), 100 mg of raffinose in the positive control group (R), and none of these in the blank group (C), respectively. The infant feces were mixed with Brain-Heart Extraction Liquid medium at the ratio of 3:7 (*w/w*) and then injected into the portioned medium at the ratio of 1:9 (*v/v*). The fermentation was conducted in triplicate at 37 °C for 48 h. AnaeroPack system (Mitsubishi Gas Chemical Co., Inc., Tokyo, Japan) was used to establish the anaerobic environment.

### 2.7. Determination of pH Value and SCFAs Content during In Vitro Fermentation

In light of the description of Chen et al. [20], the sampling procedure for pH measurement and SCFA analysis was established with minor modifications. After centrifugation at 1000× *g* rpm for 2 min, the supernatant was obtained for pH measurement and SCFA analysis, while the precipitate was for 16S rRNA gene amplicon sequencing. Abbreviations are applied to distinguish samples, using the acronym of group name and sampling interval. For example, the sample obtained in group R after 24 h of fermentation is referred to as “R24”.

Based on a headspace solid-phase microextraction (SPME) gas chromatography-mass spectrometric (GC-MS) method, SCFA analysis was performed according to Chen et al. [20] with several adjustments. One millimeter of supernatant was mixed with 105 µL of 6 M HCl and 10 µL of 20 ppm 2-methyl valeric acid (internal standard). Followed by thermal desorption in GC-MS, the absorption was accomplished by incubating the SPME fiber (DVB/CAR/PDMS) at 40 °C for 30 min, which was coated on a needle (Supelco Inc., Bellefonte, PA, USA). All external standards of acetic acid, propionic acid, isobutyric acid, butyric acid, isovaleric acid, and valeric acid were diluted from 10 to 1000 ppm. Gas chromatography-mass spectrometry analysis was conducted using an Agilent 7890B gas chromatograph, which was equipped with an Agilent 7000D mass spectrometry detector (Agilent Technologies Inc., Palo Alto, CA, USA). Chromatographic separation was achieved using an Agilent DB-WAX UI column (30 m × 0.25 µm× 0.25 µm) (Agilent Technologies Inc., Palo Alto, CA, USA). The splitless mode was selected for sample injection, and nitrogen (flow at 1 mL/min) was chosen as the carrier gas. The temperature of the gas chromatograph oven was scheduled as followed: 0–10 min, 50–100 °C, 5 °C/min; 10–14 min, 100–120 °C, 5 °C/min; 14–39 min, 120–150 °C, 2 °C/min; 39–56 min, 150–220 °C, 10 °C/min. The temperatures of both inlet and ion source were 250 °C. In a full scan mode of which the rate was 4.45 scans/s, electron impact mass spectra were recorded at an ionization voltage of 70 eV and an emission current of 35 μA. NIST search 2.3.0 software was used for data acquisition and processing. The contents of SCFAs were expressed as µmol/mL fermented broth.

### 2.8. Microbiome Analysis

DNA extraction, PCR amplification, Illumina MiSeq sequencing, and data processing were performed as described by Yun et al. [21], with minor modifications. Each PCR reaction system included 12.5 μL of 2× Taq Plus Master Mix, 3 μL of BSA (2 ng/μL), 1 μL of forward primer (5 μM), 1 μL of reverse primer (5 μM), 2 μL of DNA template, and 5.5 μL of double-distilled water. The raw data of 16S rRNA gene sequencing were deposited into the NCBI Sequence Read Archive (SRA) database (Accession Number: PRJNA782826).

### 2.9. Statistical Analysis

Data were statistically analyzed using R (http://www.r-project.org/ (accessed on 29 October 2021)) and SPSS 17.0 software (SPSS Inc., Chicago, IL, USA). Free platforms (http://www.bioinformatics.com.cn (accessed on 26 November 2021) and https://wkomics.omicsolution.com/wkomics/main/ (accessed on 26 November 2021)) were used during data analysis and figure plotting. Means comparison was carried out by Duncan’s multiple range test after one-way ANOVA. Differences were considered significant when *p* < 0.05. The correlations between SCFAs contents and microorganisms were revealed using Spearman’s correlation analysis.

## 3. Results

### 3.1. Amino Acid Composition and Peptide Profiling

The WPH was rich in Glu > Leu > Asp > Lys > Thr > Val > Ile (Figure 1). The number of essential amino acids occupied 45.09% of total amino acids (EAA/TAA), and the ratio of essential amino acids to non-essential amino acids (EAA/NEAA) was 0.82. The results of label-free quantitative proteomics showed that the peptides were mainly derived from β-lactoglobulin (59.62%), in which IPAVFK (residues 94~99) was the top-ranked peptide according to the relative abundance.

### 3.2. Growth of Lactobacilli and Bifidobacteria with WPH as Selectable Nitrogen Source

The effect of WPH on probiotic growth differed among different strains (Figure 2). All groups of LGG, BB-12, and Bi-07 reached a high cell density (OD_600_ > 1.4) during 24 h culture, regardless of the supplementation of WPH. However, the addition of 5% WPH appeared to be conducive to improving the maximum OD_600_ density or the proliferation rate of probiotics, especially for *L. acidophilus* NCFM. In the presence of 5% WPH, NCFM not only achieved an OD_600_ over 1.0 at the endpoint of the growth curve but also gained a significantly (*p* < 0.05) higher multiplication rate than that in the control group. However, the beneficial effect of WPH was not significant when observing Bi-07.

### 3.3. Changes in Molecular Weight Distribution after Digestion

The proportion of fraction <0.5 kDa increased from 27.80% to 39.36% after in vitro digestion (Figure 3). In contrast, the percentage of the fraction >10 kDa decreased from 27.36% to 11.83%. It displayed the obvious degradation of macromolecules into oligopeptides during digestion.

### 3.4. The pH Curve of Fermented Broth

During the first 6 h, the decrease in pH value already occurred in both groups L and H (Figure 4). Then, from 6 h to 12 h, a sharp decrease was observed in all groups, which was related to the production of SCFAs. Compared with group C, the reduction plummeted in groups L and H. The terminative pH of C was remarkably higher than L and H. The mild increase in group H (18–24 h) could be associated with the alkaline metabolites from peptides, such as ammonia. The diminishment of pH indicated the valid utilization of DWP by microorganisms.

### 3.5. Changes in SCFAs Formation during Fermentation

The content of SCFAs continuously accrued as the fermentation persisted (Table 1). The effect of DWP on SCFAs formation was significant for some acids, particularly from 12 h to 24 h. Only acetic acid in groups L and H attain a high concentration over 1 mol/L fermented broth. According to the productions in the terminal stage, the rank of SCFAs was acetic > butyric > propionic > iso-butyric > valeric acids. The surge of acetic acid in the second 6 h interval (6–12 h) could probably clarify the sharp decrease in pH (Figure 4) in part. Different from acetic and propionic acid, the significant increment of butyric acid was only observed in group R during the first quarter (0–12 h) of fermentation. Iso-butyric acid was not detected during the first 24 h in group R. As for iso-butyric and valeric acid, DWP noticeably (*p* < 0.05) showed a promotion in the synthesis and release of them. Intriguingly, none of the iso-valeric acids was detected in all samples during fermentation.

### 3.6. Effects of DWP on Fecal Microbiota

#### 3.6.1. Alpha and Beta Diversity Analysis

A total of 1,513,892 optimized effective sequences were obtained in microbiome analysis. The coverage of OTU at 1.00 indicated sufficient representativity for all samples (Table 2). Chao1 and observed species were used for the estimation of species richness, while Shannon and Simpson were used for the estimation of species diversity. Contrary to Shannon, the lower the Simpson value was, the higher the species diversity of the sample was. The species diversity actually covers species’ richness and species’ evenness. The significant difference (*p* < 0.05) in index values between C12 and H12 signified the noteworthy effects of DWP on fecal microbiota. Microbiota from samples in groups L and H gained a better species diversity than those in groups C and R during fermentation. More details were supplemented as Appendix A.

The principal components of PC1, PC2, and PC3 contributed 52.26%, 32.78%, and 4.84%, respectively (Figure 5A). In the graph of NMDS (Figure 5B), only samples of group C occupied the second and fourth quadrants. Almost all samples of groups L and H were in the third quadrant, whereas all of R were in the first quadrant. The separation of H24 (or L24) from C24 (or R24) was distinct in both PCoA and NMDS plots.

#### 3.6.2. Taxonomic Analysis

To figure out how DWP affects the fecal microbiota, a taxonomic analysis was carried out. In terms of the relative abundance as illustrated in Figure 5C, the dominant phyla of C0 comprised Firmicutes (67.12%), Actinobacteria (15.89%), Bacteroidota (9.53%), and Proteobacteria (7.44%). The relative ratio of Actinobacteriota and Bacteroidota significantly decreased during fermentation. However, Bacteroidota in both L12 and H24 turned out to have a higher relative abundance than the amount in C0. H24 also had a higher relative abundance of Proteobacteria than other samples. However, the relative abundance of Firmicutes in groups L and H was lower than that in groups C and R.

Meanwhile, the dominant genera of C0 contained *Roseburia* (16.10%), *Bifidobacterium* (15.64%), *Veillonella* (13.14%), *Blautia* (12.70%), *Bacteroides* (9.48%), *Escherichia-Shigella* (6.94%), and *Ruminococcus torques group* (6.20%). Conspicuous diminutions were observed at the relative abundance of *Blautia*, *Ruminococcus torques group*, *Roseburia*, *Veillonella*, *Bifidobacterium*, and *Fusicatenibacter* during fermentation, while expansions were observed on *Escherichia-Shigella* and *Paraclostridium*. A huge augmentation, followed by a later shrinkage, was detected at the relative abundance of both *Enterococcus* and *Clostridium sensu stricto 1*. There were always fewer percentages of *Enterococcus* and *Clostridium sensu stricto 1* to be observed in samples from groups L or H, instead of groups C and R. Conversely, more percentages of *Escherichia-Shigella* were found in samples from L and H. It was also noteworthy that *Bacteroides* became a stronger dominant genus in L24 and H24 after 24-h fermentation (from 9.48% in C0 to 14.39% in L24 and 22.85% in H24, respectively), even though the relative abundance in other samples did not exceed 1.00%. *Paraclostridium* was converted into a new dominant genus in L12, H12, C24, L24, and H24. Results of linear discriminant analysis effect size (LEfSe) analysis are supplied as Appendix A. Specifically, members of *Lactococcus*, *Escherichia Shigella*, *Paraclostridium*, *Eggerthella*, *Pygmaiobacter*, *Bacteroides*, *Anoxybacillus*, and *Clostridioides* were the differentially abundant genera (LDA score over 3) in groups L and H.

### 3.7. Correlation Analysis of SCFAs with Fecal Microbiota at the Genus Level

As presented in Figure 6, *Clostridium sensu stricto 1* and *Enterococcus* showed positive correlations with acetic acid changes, despite negative correlations with propionic acid, butyric acid, and valeric acid changes. Similarly, *Blautia* showed a weaker positive correlation with acetic acid, but a stronger negative correlation with valeric acid. On the contrary, *Bacteroides*, *Pygmaiobacter*, and *Clostridioides* negatively correlated with acetic acid changes, while they were positively correlated with other SCFAs. Meanwhile, *Paraclostridium* also had a positive correlation with propionic acid and iso−butyric acid changes, but a weak negative correlation with acetic acid. Furthermore, *Ruminococcus gnavus group*, *Escherichia Shigella*, *Streptococcus*, and *Erysipelatoclostridium* seemed to be uncorrelated with the variations in SCFAs.

## 4. Discussion

Compared with several common protein sources that comprised collagen, zein, sorghum flour, wheat bran cereals, peanuts, black beans, and pigeon peas, WP showed thorough protein hydrolysis during in vitro digestion and the highest EAA/NEAA ratio [22]. In many reports, WP often served as a good source of branched-chain amino acids (BCAA) and tryptophan. Some clinical evidence has suggested that WP potentially has beneficial effects on the immune system and muscle synthesis [23,24], which may be healthful for infants. It is hypothesized that the high content of BCAA may promote protein synthesis in muscles; meanwhile, the high amount of lysine and arginine could stimulate both anabolic hormone and growth hormone [25]. In this study, the WPH still retains abundant contents of BCAA (leucine, valine, and isoleucine), lysine, and arginine. A certain amount of tryptophan was also determined in peptides such as the 35th tryptophan residue in β-lactoglobulin. The high ratio of EAA/NEAA at 0.82 signifies a high-quality protein according to the FAO/WHO, which defines high-quality proteins as EAA/NEAA ≥ 0.6 [26]. The ratio of fraction <0.5 kDa in WPH increased to 39.36% during gastrointestinal digestion, indicating the facilitation in peptide release and absorption. From these points of view, WPH deserves to be considered as an optional supplement to essential amino acids in IF. Moreover, gut bacteria were found to preferentially assimilate and ferment peptides over amino acids during culture-based experiments, which is likely related to the energetical efficiency [27]. Thus, it is necessary to evaluate the effects of WPH on the infant gut microbiome.

Firstly, five strains from four probiotic species were used for 24 h separate culture, in which WPH occupied 3% or 5% of the nitrogen source. In comparison with cultured bifidobacteria, the prebiotic effect was more significant on the growth of lactobacilli, especially on *Lactobacillus acidophilus* NCFM. The diversity of pathways among different bacterial species at the strain level was potentially related to these observations. Known as a safe supplement that can reach the gut after oral administration, NCFM also exhibited the potential to help the digestion of lactose, support immune function, and alleviate the slow transit of stools [28]. To obtain such probiotics or beneficial balance, numerous discussions have focused on the early formation of gut microbiota, particularly during the perinatal period. Among perinatal factors, delivery mode, drug administration, and many other conditions have been taken into consideration, while neonatal nutrition has drawn special attention [29]. Breastfeeding can not only offer oligosaccharides to promote the growth of *Bifidobacterium* species [30], but also allows the transmission of a special lactobiome to conduct positive modulation [29]. Some specific bioactive compounds have been supplemented concurrently with IFs to meet infant needs, such as α-Lactalbumin, probiotics, prebiotics, etc. [31]. The symbiotic combination with NCFM could be a more effective way to utilize WPH.

Then, to better determine how WPH affects gut microbiota, the procedure combined in vitro digestion and 48 h gut fecal fermentation was established to mimic colonic fermentation. In consonance with Moya’s findings on whey milk [32], the rank of SCFAs after 48 h fecal fermentation was “acetate > propionate > butyrate” acids. The total amount of acetate, butyrate, and propionic was dominant in SCFAs (over 90%). Actually, SCFAs in the human gut primarily originate from the fermentation of nondigestible carbohydrates, which amounts to a production of SCFAs of 0.24~0.38 kg body weight^−1^ h^−1^ [33]. Although colonic protein fermentation also offered SCFAs, it is more preferable for gut microbiota to ferment carbohydrates for ATP, which can facilitate the biosynthesis of proteins using nitrogenous substrates [12]. This may explain the sharpest decrease in pH (Figure 4) induced by raffinose, which was in agreement with another in vitro fermentation reported before [20]. However, the SCFAs in group R were lower than those in groups L or H. Group L or H also had an earlier detection of isobutyrate, which was produced from amino acids (particularly valine) instead of carbohydrates [34]. On the one hand, it probably indicated the different pathways to generate SCFAs. There might be more organic acids (such as formate, lactate, and succinate) derived from raffinose in group R, while in groups L and H there may be more alkaline products due to the oxidative deamination. On the other hand, it implied that proteolytic fermentation of DWP could supply more counts of SCFAs that were accompanied by an equal reduction in pH.

Concerning the microbiota in group H, a distinction was the lowest ratio of Firmicutes/Bacteroidetes in H24. It was reported that the ratio of “Firmicutes: Bacteroidetes” in stools was higher among some obese people [33,35]. Generally, butyrate is regarded as the primary metabolic end product of Firmicutes, while Bacteroidetes mainly produce acetate and propionate. So, the composition of SCFAs could be modulated by DWP. In addition, compared with groups C and R, DWP caused a higher relative abundance of *Proteobacteria*, *Bacteroides*, and *Streptococcus*. All of them belong to the most abundant amino acid fermenting bacteria in the human small intestine [36]. Known as the candidate for attenuating inflammation by regulating lymphocytes and cytokine expression, the *Bacteroides* genus has the strong polysaccharides degradation system to participate in SCFAs production (mostly in the form of acetate and propionate) [37]. *Bacteroides* took over the dominance of Firmicutes and *Bifidobacterium* as infants grow up, which revealed its important role in the maturation of the infant gut [38]. The up-regulated abundance of *Bacteroides* was also reported in obese mice that received a positive modulatory effect of conjugated linoleic acid, which facilitated the prevention of obesity-related metabolic disorders [39]. In another study of obese mice, bovine α-lactalbumin hydrolysates increased the ratio of Bacteroidetes/Firmicutes and decreased the level of lipopolysaccharides in feces and serum, inducing the amelioration of systematic inflammation and endotoxemia [40]. Analogously, in a randomized controlled double-blind study among athletes, a protein supplement (whey isolate and beef hydrolysate) induced a higher abundance of Bacteroidetes phylum [41]. As a result, WPH had the potential to improve energy balance and metabolism by reshaping the structure of gut microbiota.

## 5. Conclusions

In summary, the WPH enriched in essential amino acids significantly promoted the growth of *Lactobacillus acidophilus* NCFM. It implied the potential prebiotic property of WPH as a nutritional supplement. Furthermore, the addition of WPH induced a sharper reduction in pH and a higher production of SCFAs than the blank group during in vitro fecal fermentation. A lower ratio of Firmicutes/Bacteroidetes was found in the microbiota, indicating a possible improvement in the energy balance and metabolism of the host.

## Figures and Tables

**Figure 1 nutrients-14-01374-f001:**
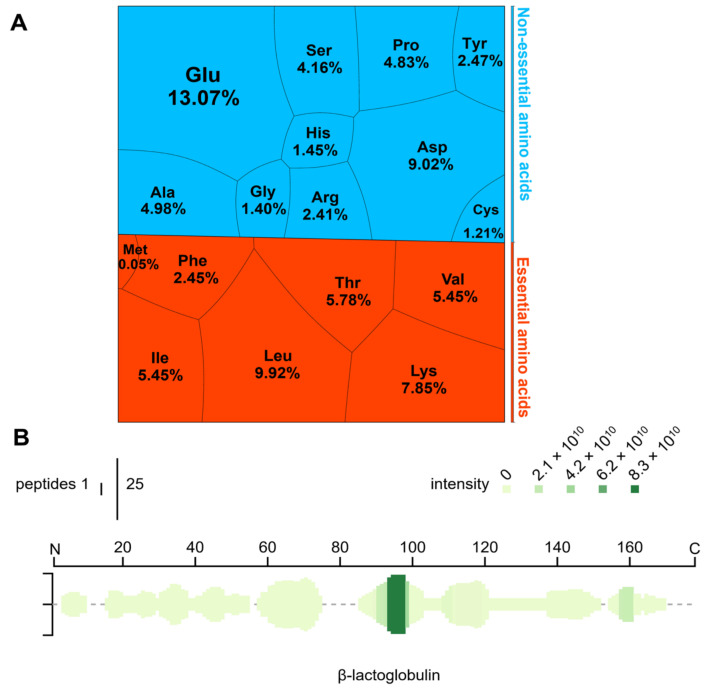
Basic information on the proteinaceous components of whey protein hydrolysate (WPH): (**A**) The Voronoi treemap is based on the amino acid composition. (**B**) Peptide profile of β-lactoglobulin in WPH. Gly, Glycine; Ala, Alanine; Val, Valine; Leu, Leucine; Ile, Isoleucine; Met, Methionine; Phe, Phenylalanine; Trp, Tryptophan; Pro, Proline; Ser, Serine; Thr, Threonine; Cys, Cysteine; Tyr, Tyrosine; Asn, Asparagine; Gln, Glutamine; Asp, Aspartic acid; Glu, Glutamic acid; Lys, lysine; Arg, Arginine; His, Histidine.

**Figure 2 nutrients-14-01374-f002:**
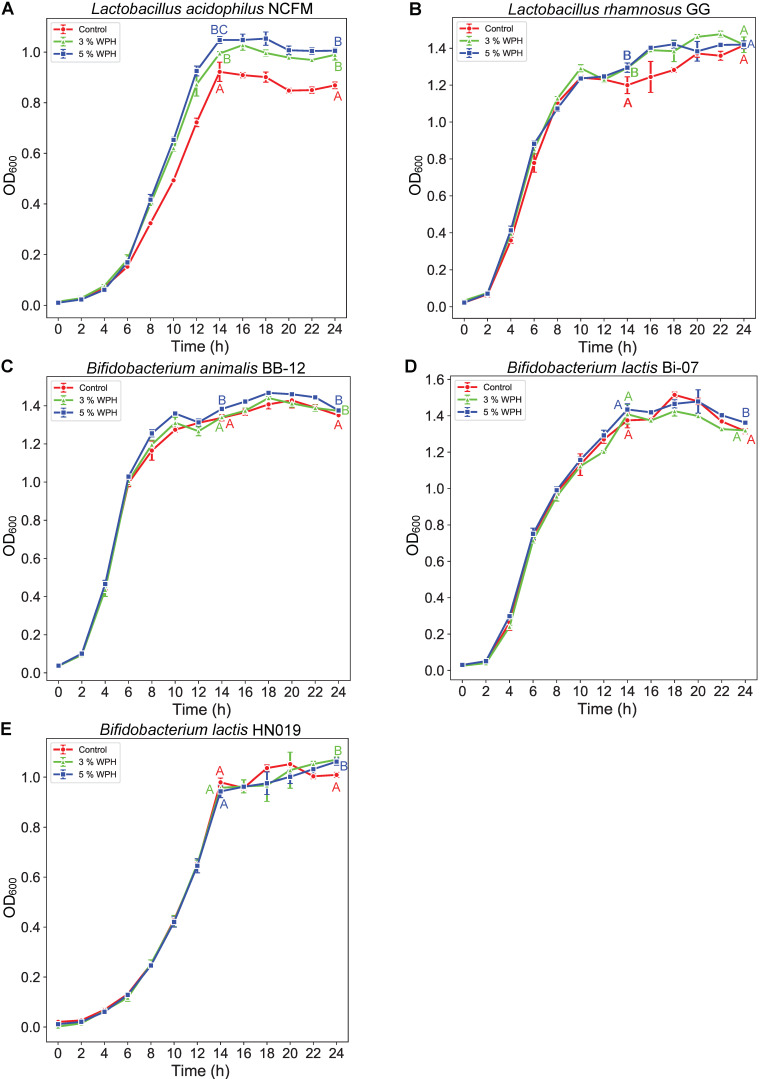
Growth curves of cultivated probiotics: (**A**) *Lactobacillus acidophilus* NCFM, (**B**) *Lactobacillus rhamnosus* GG, (**C**) *Bifidobacterium animalis* BB-12, (**D**) *Bifidobacterium lactis* Bi-07, (**E**) *Bifidobacterium lactis* HN019. Different capital superscript letters indicate significant differences (*p* < 0.05) for OD_600_ value in different samples (at the time point of 14 h and 24 h, respectively).

**Figure 3 nutrients-14-01374-f003:**
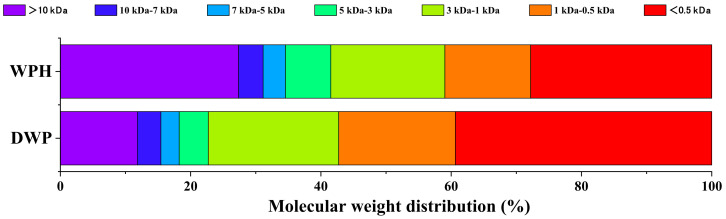
The molecular weight distribution of whey protein hydrolysate (WPH) and its digestive products (DWP).

**Figure 4 nutrients-14-01374-f004:**
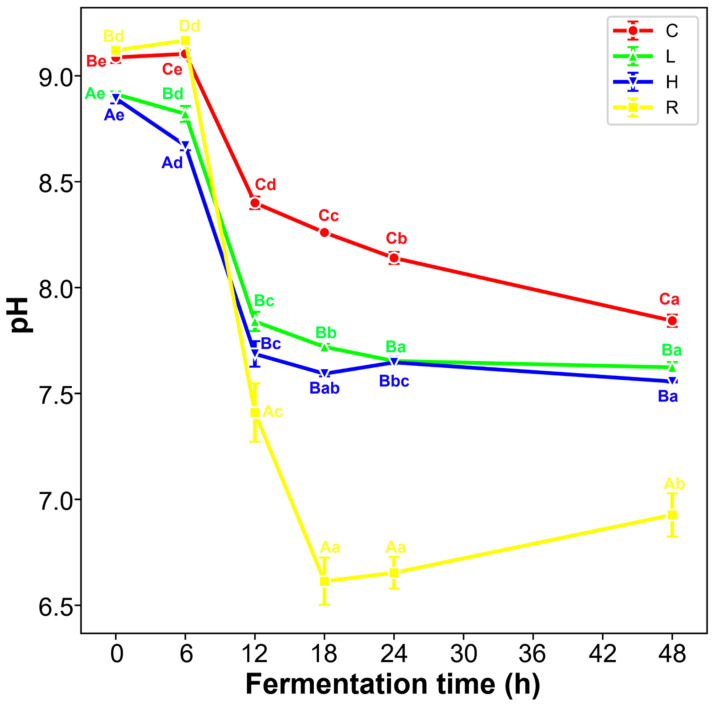
The pH changes in infant feces during fermentation. Different capital superscript letters indicate significant differences for each pH value in different groups, while different lowercase superscript letters indicate significant differences (*p* < 0.05) for each pH value on different dates of sampling. C, blank group; L, low-dose group; H, high-dose group; R, positive control group.

**Figure 5 nutrients-14-01374-f005:**
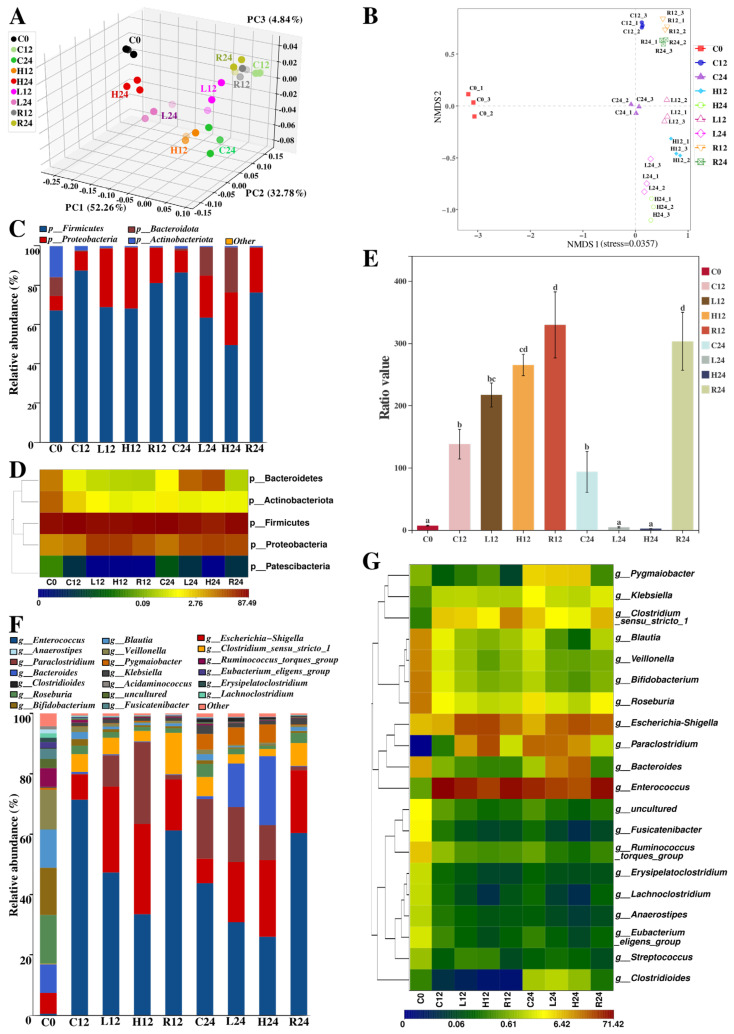
Beta diversity analysis and taxonomic analysis of fecal microbiota: (**A**) The principal coordinates analysis (PCoA) is based on weighted Unifrac distance. (**B**) The nonmetric multidimensional scaling (NMDS). (**C**) Microbial taxa composition at the phylum level. (**D**) Hierarchical cluster analysis of main OTUs at the phylum level. (**E**) The ratios of Firmicutes/Bacteroidetes. Different lowercase superscript letters indicate significant differences (*p* < 0.05) in different samples. (**F**) Microbial taxa composition at the genus level. (**G**) Hierarchical cluster analysis of main OTUs at the genus level. PC, Principal component; C, blank group; L, low-dose group; H, high-dose group; R, positive control group.

**Figure 6 nutrients-14-01374-f006:**
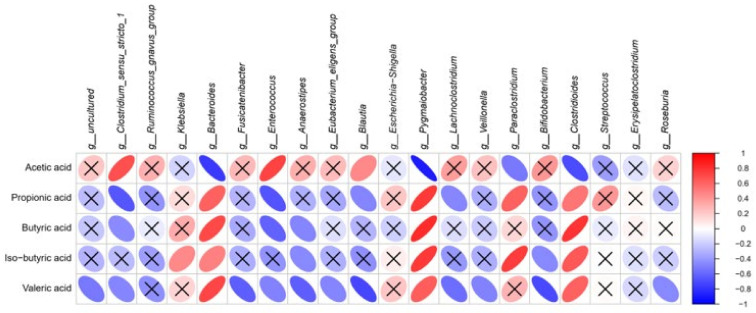
Spearman’s correlation analysis of SCFAs contents with microorganisms. × The insignificant correlation (*p* > 0.05); g, genus.

**Table 1 nutrients-14-01374-t001:** Production content of SCFAs during fermentation (µmol/mL fermented broth).

SCFA	Sample	0 h	6 h	12 h	18 h	24 h	48 h
Acetic acid	C	96.04 ± 5.83 ^A,a^	110.35 ± 3.34 ^A,a^	354.65 ± 70.06 ^A,b^	480.95 ± 9.26 ^A,c^	675.90 ± 68.17 ^A,d^	704.89 ± 56.67 ^A,d^
	L	101.52 ± 9.92 ^A,a^	129.76 ± 5.15 ^B,a^	467.93 ± 6.30 ^B,b^	679.48 ± 19.82 ^B,c^	966.73 ± 72.05 ^B,d^	1068.40 ± 91.21 ^C,d^
	H	98.25 ± 9.74 ^A,a^	154.25 ± 7.88 ^C,a^	476.35 ± 1.94 ^B,b^	690.45 ± 17.61 ^B,c^	1029.74 ± 44.60 ^B,d^	978.99 ± 161.96 ^BC,d^
	R	96.05 ± 6.09 ^A,a^	111.97 ± 11.91 ^AB,a^	450.27 ± 58.52 ^AB,b^	599.20 ± 99.70 ^AB,c^	592.45 ± 66.52 ^A,c^	810.98 ± 51.61 ^AB,d^
Propionic acid	C	7.97 ± 0.66 ^B,a^	5.62 ± 0.50 ^A,a^	17.06 ± 2.96 ^AB,a^	48.09 ± 4.70 ^B,b^	91.01 ± 19.42 ^B,c^	126.48 ± 4.92 ^B,d^
	L	6.91 ± 1.16 ^AB,a^	7.61 ± 1.07 ^B,a^	35.89 ± 2.72 ^BC,b^	89.10 ± 2.10 ^C,c^	144.74 ± 7.18 ^C,d^	189.79 ± 3.30 ^C,e^
	H	5.91 ± 0.82 ^AB,a^	8.03 ± 0.15 ^B,a^	63.96 ± 24.10 ^C,b^	100.27 ± 2.73 ^D,c^	162.49 ± 0.99 ^C,d^	171.81 ± 29.20 ^C,d^
	R	4.63 ± 1.53 ^A,a^	6.25 ± 1.07 ^AB,a^	6.40 ± 2.16 ^A,a^	10.56 ± 3.22 ^A,a^	12.74 ± 3.69 ^A,a^	43.61 ± 2.84 ^A,a^
Butyric acid	C	4.20 ± 0.99 ^B,a^	2.77 ± 0.10 ^A,a^	3.96 ± 1.10 ^A,a^	6.94 ± 0.31 ^A,a^	39.36 ± 3.21 ^B,b^	180.29 ± 4.62 ^B,c^
	L	3.15 ± 0.52 ^AB,a^	4.51 ± 1.14 ^B,a^	5.47 ± 0.15 ^A,a^	41.08 ± 4.21 ^B,b^	101.31 ± 3.81 ^C,c^	201.53 ± 10.30 ^BC,d^
	H	2.73 ± 0.35 ^AB,a^	3.66 ± 0.11 ^AB,a^	8.83 ± 4.31 ^A,a^	66.53 ± 4.22 ^C,b^	123.37 ± 9.59 ^D,c^	208.20 ± 19.41 ^C,d^
	R	2.17 ± 0.52 ^A,a^	2.76 ± 0.53 ^A,a^	7.59 ± 1.94 ^A,b^	11.41 ± 2.26 ^A,c^	19.36 ± 0.92 ^A,d^	144.03 ± 0.25 ^A,e^
Iso-butyric acid	C	ND	ND	ND	18.83 ± 5.99 ^A,a^	51.38 ± 4.08 ^A,b^	91.35 ± 4.30 ^B,c^
	L	ND	ND	ND	33.31 ± 2.60 ^B,a^	60.29 ± 3.27 ^B,b^	120.16 ± 0.44 ^C,c^
	H	ND	ND	6.48 ± 0.61 ^A,a^	34.27 ± 0.42 ^B,b^	57.07 ± 1.02 ^AB,c^	98.58 ± 14.28 ^B,d^
	R	ND	ND	ND	ND	ND	22.92 ± 3.65 ^A,a^
Valeric acid	C	ND	ND	ND	ND	ND	4.50 ± 0.50 ^B,a^
	L	ND	ND	ND	ND	1.13 ± 0.19 ^A,a^	8.69 ± 0.34 ^C,b^
	H	ND	ND	ND	ND	1.26 ± 0.09 ^A,a^	8.30 ± 0.89 ^C,b^
	R	ND	ND	ND	ND	ND	2.30 ± 1.33 ^A,a^
Iso-valeric acid	C	ND	ND	ND	ND	ND	ND
	L	ND	ND	ND	ND	ND	ND
	H	ND	ND	ND	ND	ND	ND
	R	ND	ND	ND	ND	ND	ND

Different capital superscript letters indicate significant differences (*p* < 0.05) for each SCFA in different groups; Different lowercase superscript letters indicate significant differences (*p* < 0.05) for each SCFA in different dates of sampling; C, blank group; L, low-dose group; H, high-dose group; R, positive control group; ND, not detected; *n* = 3, $\bar{x}$ ± SD.

**Table 2 nutrients-14-01374-t002:** Alpha diversity estimation of fecal microbiota during fermentation.

Sample ID	Observed Species	Chao1	Shannon	Simpson	Coverage
C0	82.00 ± 0.82 ^A^	88.18 ± 3.48 ^A^	4.27 ± 0.05 ^A^	0.08 ± 0.00 ^E^	1.00 ± 0.00
C12	67.67 ± 0.94 ^BC^	82.27 ± 9.38 ^AB^	2.29 ± 0.07 ^D^	0.41 ± 0.01 ^A^	1.00 ± 0.00
C24	70.67 ± 2.05 ^B^	82.54 ± 1.15 ^AB^	3.17 ± 0.19 ^B^	0.20 ± 0.02 ^C^	1.00 ± 0.00
L12	66.33 ± 2.49 ^C^	86.98 ± 9.17 ^A^	2.63 ± 0.04 ^C^	0.24 ± 0.01 ^B^	1.00 ± 0.00
L24	67.00 ± 0.82 ^C^	78.26 ± 7.54 ^ABC^	3.31 ± 0.07 ^B^	0.15 ± 0.01 ^D^	1.00 ± 0.00
H12	57.33 ± 1.70 ^E^	64.56 ± 4.28 ^C^	2.51 ± 0.08 ^C^	0.24 ± 0.01 ^B^	1.00 ± 0.00
H24	61.67 ± 0.47 ^D^	67.29 ± 5.45 ^BC^	3.25 ± 0.00 ^B^	0.15 ± 0.00 ^D^	1.00 ± 0.00
R12	60.33 ± 2.05 ^DE^	71.33 ± 3.92 ^ABC^	1.99 ± 0.07 ^E^	0.40 ± 0.02 ^A^	1.00 ± 0.00
R24	62.67 ± 0.47 ^D^	75.83 ± 12.22 ^ABC^	2.13 ± 0.15 ^DE^	0.39 ± 0.03 ^A^	1.00 ± 0.00

The different capital superscript letters indicate significant differences (*p* < 0.05) for each index in different samples; R24, the sample obtained in group R after 24 h of fermentation; C, blank group; L, low-dose group; H, high-dose group; R, positive control group; *n* = 3, $\bar{x}$ ± SD.

## Data Availability

The raw data of 16S rRNA gene sequencing are available from NCBI Sequence Read Archive (SRA) database under the accession number PRJNA782826.

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
