# Peer review of "In Vitro Gut Fermentation of Whey Protein Hydrolysate: An Evaluation of Its Potential Modulation on Infant Gut Microbiome"

_nutrients, 2022, doi:10.3390/nu14071374_

Round 1

Reviewer 1 Report

Chunsong Feng et al. sent an article entitled „In Vitro Gut Fermentation of Whey Protein Hydrolysate: An Evaluation of Its Potential Modulation on Infant Gut Microbiome” to Nutrients.

The authors evaluated the effect of whey protein hydrolysate (WPH) on the infant gut microbiome, which was more variable than that of adults. Colonic fermentation was simulated through a static digestion model and fecal culture fermentation, using feces from normal infants aged from 1-3 years old. During in vitro gut fermentation, measurements of short-chain fatty acids (SCFA) concentrations and 16S rRNA amplicon sequencing were performed.

Besides the decline of pH in fermentation, the addition of WPH induced a significant increase in the SCFA production and also the relative abundance of Proteobacteria, Bacteroides, and Streptococcus (p <  0.05).  The stimulation effect of WPH on the probiotics (particularly Lactobacillus acidophilus NCFM) was detected during cultivation which implied the prebiotic potential also.

They concluded that WPH is a valuable dietary supplement with not only enriched resources of essential amino acids but also has the potential to restore the infant gut microbiome.

The prevalence of allergies to cows’ milk in infants is high. Whey protein hydrolysate has been commercially developed as a major kind of hypoallergenic dairy product. Until recently only a few studies evaluated the potential effect of hydrolyzed whey protein on the infant gut microbiome.

The experimental setup, the applied methods were carefully selected and properly detailed in the text. Figures, tables are convincing, the conclusions are appropriate.

As a reviewer I would like to suggest:

  1. In an article, the weakness of the study usually is a part of the discussion. In the present study feces from normal infants aged from 1- to 3 years were used. They characterized the samples as “normal”. It is well known that during the perinatal period, neonatal microbiota seems to be influenced by the mode of delivery, drug administration, and many other conditions. Special attention must be reserved for early neonatal nutrition because breastfeeding allows the transmission of a specific and unique lactobiome able to modulate and positively affect the neonatal gut microbiota. Therefore, the perinatal history should be taken into consideration.

(Coscia, A.; Bardanzellu, F.; Caboni, E.; Fanos, V.; Peroni, D.G. When a Neonate Is Born, So Is a Microbiota. Life 2021, 11, 148).

https://doi.org/10.3390/life11020148

  1. In this paper, numerous sentences begin with “And”, which many times sounds to be inappropriate.

Author Response

The authors are very grateful to the editors and reviewers for seriously and quickly proposing insightful advice to improve the quality of the manuscript.

Response to the comments of Reviewer #1:

Reviewer #1: Chunsong Feng et al. sent an article entitled „In Vitro Gut Fermentation of Whey Protein Hydrolysate: An Evaluation of Its Potential Modulation on Infant Gut Microbiome” to Nutrients. The authors evaluated the effect of whey protein hydrolysate (WPH) on the infant gut microbiome, which was more variable than that of adults. Colonic fermentation was simulated through a static digestion model and fecal culture fermentation, using feces from normal infants aged from 1-3 years old. During in vitro gut fermentation, measurements of short-chain fatty acids (SCFA) concentrations and 16S rRNA amplicon sequencing were performed. Besides the decline of pH in fermentation, the addition of WPH induced a significant increase in the SCFA production and also the relative abundance of Proteobacteria, Bacteroides, and Streptococcus (p <  0.05).  The stimulation effect of WPH on the probiotics (particularly Lactobacillus acidophilus NCFM) was detected during cultivation which implied the prebiotic potential also. They concluded that WPH is a valuable dietary supplement with not only enriched resources of essential amino acids but also has the potential to restore the infant gut microbiome. The prevalence of allergies to cows’ milk in infants is high. Whey protein hydrolysate has been commercially developed as a major kind of hypoallergenic dairy product. Until recently only a few studies evaluated the potential effect of hydrolyzed whey protein on the infant gut microbiome. The experimental setup, the applied methods were carefully selected and properly detailed in the text. Figures, tables are convincing, the conclusions are appropriate.

As a reviewer I would like to suggest:

  1. In an article, the weakness of the study usually is a part of the discussion. In the present study feces from normal infants aged from 1- to 3 years were used. They characterized the samples as “normal”. It is well known that during the perinatal period, neonatal microbiota seems to be influenced by the mode of delivery, drug administration, and many other conditions. Special attention must be reserved for early neonatal nutrition because breastfeeding allows the transmission of a specific and unique lactobiome able to modulate and positively affect the neonatal gut microbiota. Therefore, the perinatal history should be taken into consideration.

(Coscia, A.; Bardanzellu, F.; Caboni, E.; Fanos, V.; Peroni, D.G. When a Neonate Is Born, So Is a Microbiota. Life 2021, 11, 148).https://doi.org/10.3390/life11020148

Response: Thanks for your comments. We agree with the reviewer that neonatal microbiota the perinatal history is worthy of being considered. The corresponding discussion has been supplemented (line 329~335).

  1. In this paper, numerous sentences begin with “And”, which many times sounds to be

Response:  Thanks for your useful reminding. Several “And” have been deleted (line 52, 87, 171, 216, 258, 268, 274, 315, 335, 342, 368, and 385).

Reviewer 2 Report

In the manuscript titled "In Vitro Gut Fermentation of Whey Protein Hydrolysate: An Evaluation of Its Potential Modulation on Infant Gut Microbiome" by Chunsong Feng and colleagues, they have reported that the addition of whey protein hydrolysate induced a significant increase in the SCFA production and also the relative abundance of Proteobacteria, Bacteroides, and Streptococcus. The lower ratio of Firmicutes/ Bacteroidetes in whey protein hydrolysate-supplemented samples indicated the positive modulation of whey protein hydrolysate on the gut microbiota, which could benefit the energy balance and metabolism of infants. The stimulation effect of whey protein hydrolysate on the probiotics (particularly Lactobacillus acidophilus NCFM) during cultivation implied the prebiotic potential as well. I have a few comments regarding the present original manuscript

-Maybe the introduction requires another way to introduce the whey protein hydrolysate, such as health-related and why this protein is important to human health

-The infant stool samples were used as controls, how the authors have estimated that the 10 samples were similar, which control or controls were assessed?

-Important information in the microbiome analysis section, how the samples were treated, and which programs were used to obtain and show the results

-Also, statistical analyses needs more information, which program was used to perform the correlations figures

-Some typos have been found in the discussion, especially the italics of genera or phyla, depending also on the journal guidelines.  

Author Response

The authors are very grateful to the editors and reviewers for seriously and quickly proposing insightful advice to improve the quality of the manuscript.

Response to the comments of Reviewer #2:

Reviewer #2: In the manuscript titled "In Vitro Gut Fermentation of Whey Protein Hydrolysate: An Evaluation of Its Potential Modulation on Infant Gut Microbiome" by Chunsong Feng and colleagues, they have reported that the addition of whey protein hydrolysate induced a significant increase in the SCFA production and also the relative abundance of Proteobacteria, Bacteroides, and Streptococcus. The lower ratio of Firmicutes/ Bacteroidetes in whey protein hydrolysate-supplemented samples indicated the positive modulation of whey protein hydrolysate on the gut microbiota, which could benefit the energy balance and metabolism of infants. The stimulation effect of whey protein hydrolysate on the probiotics (particularly Lactobacillus acidophilus NCFM) during cultivation implied the prebiotic potential as well. I have a few comments regarding the present original manuscript

  1. Maybe the introduction requires another way to introduce the whey protein hydrolysate, such as health-related and why this protein is important to human health?

Response:  Thanks for your valuable reminding. Matching improvements have been made in our introduction part (line 32, line 44).

  1. The infant stool samples were used as controls, how the authors have estimated that the 10 samples were similar, which control or controls were assessed?

Response:  Thanks for your comment. All of the samples were mixed uniformly before inoculation. The corresponding statement in the method section has been added (line109). One hundred microgram of raffinose in the positive control group (R), and no extra prebiotics were supplemented in the blank group (C) for fermentation.

  1. Important information in the microbiome analysis section, how the samples were treated, and which programs were used to obtain and show the results

Response:  Thanks for your comment. Our processes in the microbiome analysis section are almost the same as those of the cited research (line 151). The minor modifications are all listed in detail. And the text of the corresponding description in this section is too excessive to be displayed in this manuscript. Otherwise, the method part seems to be oversized and the manuscript could not be succinct enough. We sincerely hope it can meet the journal’s standards.

  1. Also, statistical analyses needs more information, which program was used to perform the correlations figures

Response:  Thanks for your comment. The Spearman’s correlation analysis in our study was conducted using the 'Wu Kong' platform (established on the basis of R language), which was mentioned in our initial manuscript (line 162). We have checked it again and updated the latest website.

  1. Some typos have been found in the discussion, especially the italics of genera or phyla, depending also on the journal guidelines.

Response:  Thanks for your reminding, we have revised our manuscript thoroughly. We had also found those slight differences before our submission, when browsing through “Catalogue of Life” (the database provides a consistent and up-to-date listing of all the world’s known species). In this study, sequences were taxonomically classified against SILVA 16S rRNA gene reference database release 128, using Ribosomal Database Project (RDP) Classifier tool. Those designations ( like the “Clostridium sensu stricto 1”) are used to refer to OTUs (Operational Taxonomic Units) more directly, instead of the specific names in taxonomy. Similar cases can also be found in many publications, such as the “Clostridium sensu stricto 1” in the research article named “Protective effects of polysaccharides from Atractylodes macrocephalae Koidz. against dextran sulfate sodium induced intestinal mucosal injury on mice”.

Round 2

Reviewer 2 Report

Thank you to the authors for giving me a very detailed response to my previous questions. The manuscript improved after the peer-review process. No further comments are required from my side, again thanks